# Rapid Personalised Virtual Planning and On-Demand Surgery for Acute Spinal Trauma Using 3D-Printing, Biomodelling and Patient-Specific Implant Manufacture

**DOI:** 10.3390/jpm12060997

**Published:** 2022-06-18

**Authors:** Ralph Jasper Mobbs, William C. H. Parr, Christopher Huang, Tajrian Amin

**Affiliations:** 1NeuroSpine Surgery Research Group (NSURG), Sydney, NSW 2031, Australia; w.parr@unsw.edu.au (W.C.H.P.); christopherhuang3@gmail.com (C.H.); tajamin1998@gmail.com (T.A.); 2Neuro Spine Clinic, Prince of Wales Private Hospital, Randwick, NSW 2031, Australia; 3Faculty of Medicine, University of New South Wales (UNSW), Sydney, NSW 2031, Australia; 4Surgical and Orthopaedic Research Laboratories (SORL), Prince of Wales Clinical School, University of New South Wales, Randwick, NSW 2031, Australia; 53DMorphic Pty. Ltd., Matraville, NSW 2036, Australia

**Keywords:** on-demand, personalised health care, 3D printing, virtual surgical planning

## Abstract

Three-dimensional printing is a rapidly growing field, with extensive application to orthopaedics and spinal surgery. Three-dimensional-printed (3DP) patient-specific implants (PSIs) offer multiple potential benefits over generic alternatives, with their use increasingly being described in the spinal literature. This report details a unique, emergency case of a traumatic spinal injury in a 31-year-old male, acquired rurally and treated with a 3DP PSI in a tertiary unit. With increasing design automation and process improvements, rapid, on-demand virtual surgical planning (VSP) and 3DP PSIs may present the future of orthopaedics and trauma care, enabling faster, safer, and more cost-effective patient-specific procedures.

## 1. Background

Although the application of three-dimensional printing (3DP) to medical prosthesis production is relatively new, spinal prostheses manufactured by 3DP have undergone rapid acceptance over the past five years, with 3DP devices now widely seen as a promising field. The underlying principle of 3DP is the layer-by-layer deposition of material to manufacture a three-dimensional part [1,2]. A key feature of 3DP is that complex device geometries can be manufactured as easily as simple ones. This has led to interest and application of 3DP to manufacture personalised, custom-made implants. Increasing awareness and accessibility have facilitated the increasing application of the personalised approach in clinical practice. This has included patient-specific biomodels, custom-made surgical instruments, guides, and implants, also termed patient-specific implants (PSIs), being utilised throughout the spine [2].

While traditional “Off-The-Shelf” spinal implants have required remodelling of the patient’s anatomy to fit the implant, 3DP offers the possibility of modelling the implant to fit the patient’s anatomy [3]. This may allow for improved primary stabilization, endplate loading and stress distribution, osseointegration and spinal curvature correction [3,4]. Intraoperatively, PSIs may have improved “fit” and reduced operative time, blood loss and surgical difficulty [5,6]. Prior publications have suggested that PSIs may reduce pain and complications, including neurovascular issues, stress shielding and cage subsidence [2,7].

However, these technologies face a number of challenges including design and manufacturing time and cost effectiveness; the need for specialised skills and equipment; the logistical challenges of point-of-care delivery and changing regulatory environments [2,7]. Given the nascence of the field, there is an understandable lack of highly powered supportive evidence, with much of the extant research comprising case reports, often of complex cases, and small series. Additionally, many of these reported cases have involved prolonged preoperative planning and device manufacture, meaning that these technologies have been used to address non-emergency, predominantly degenerative and tumour cases.

Herein, we present the first case of emergency traumatic spinal injury that was treated, successfully, by an on-demand PSI.

## 2. Materials and Methods

### 2.1. Case Presentation

A 31-year-old male sustained a C7 burst fracture, with an associated American Spinal Injury Association (ASIA) C spinal cord injury, following a fall directly on his head (axial loading) during a game of Australian Rules Football in a rural location; 600 km from a tertiary spinal injury unit. Clinically, the patient demonstrated a central cord syndrome with 3-4/5 power in the lower extremities and 1-2/5 power in the upper extremities (no anti-gravity power). On arrival at the peripheral hospital, a computer tomography (CT) scan was performed which demonstrated the C7 fracture and spinal canal stenosis (Figure 1). The tertiary hospital with a spinal injuries unit was contacted for transfer and ongoing management.

The CT digital imaging and communications in medicine (DICOM) images were transferred to the spinal injuries unit, where the neurosurgeon (R.J.M) and computer-assisted design engineer (W.C.H.P) determined the implant design requirements and fixation options. Virtual surgery planning (VSP) provided further design inputs allowing for the design to be finalised and printed (3DMorphic, Sydney, Australia) in Titanium alloy using an EOSM100 (Krailling, Germany) (Figure 1) A patient-specific biomodel and resin versions of the PSIs were 3DP while the patient was in transit. The patient arrived 11 h post-injury to the tertiary hospital. Surgery proceeded as per the VSP with minimal blood loss (<20 cc) via a standard anterior cervical exposure. Immediate intraoperative firm fixation was achieved with a postoperative CT scan demonstrating equivalent implant positioning as per the VSP (Figure 2).

### 2.2. Preoperative Planning

Preoperative planning was performed immediately upon receipt of the medical images (CT DICOMs) from the peripheral hospital and included: CT segmentation; 3D reconstruction (Materialise MIMICS (vs 22.0) and 3 Matic (vs 14.0) Leuven, Belgium [8,9,10]); 3DP of the patient’s C7 pathological anatomy (clear resin, Form2, Formlabs, Somerville, MA, USA); VSP and 3DP of a dynamic biomodel of the planned corpectomy in Acrylonitrile Butadiene Styrene (ABS, Stratasys, Eden Prairie, MN, USA) and PSI design (3DMCAD, 3DMorphic, Sydney, Australia).

Intraoperative goals were established preoperatively through consultation with the surgical team and the VSP process. Standard qualitative surgical goals, including decompression and fusion, with the clinical goals, namely spinal cord decompression to restore neurological function, were combined with quantitative goals, including targets for restoration of vertebral body height (distraction) and sagittal balance. Haptic biomodels allowed the surgical team to explain the procedure to the patient, plan the surgery and were used intraoperatively as an integral part of the PSI implantation. The models were interactive, in that they allowed for 3DP (photo-polymerised methacrylate, Formlabs, Somerville, MA, USA) resin cages to be trialled in the biomodel, with corresponding changes to the vertebral height and lordotic angle assessed (Figure 1).

The PSI was 3DP using direct metal laser solidification (DMLS) of Ti_6_Al_4_V (Biomedical Grade 5) titanium alloy powder (EOS, Krailling, Germany). Ti_6_Al_4_V was used due to its biocompatibility, relatively low stiffness (~110 GPa) and radiologic scatter compared to steel (316l) or Cobalt Chrome metallic alternatives [11,12,13,14]. Cages were produced at three heights and lordotic angles to provide intra-operative options for the surgical team regarding optimal fit and anatomical reconstruction. Screw trajectories and lengths were determined in the VSP, with trajectories incorporated in the PSI design via screw hole angulation.

### 2.3. Operative Technique

The anterior cervical surgical technique has been described, as well as its application using a 3DP PSI [4]. Briefly, the patient was positioned supine. Using image intensified fluoroscopy, the C7 level was confirmed prior to incision. A linear incision was made on the right of the midline. A Caspar and TrimLine retractor were used for exposure. Image intensification was used to confirm the level of pathology. The fractured C7 vertebral body was removed, and canal decompression was achieved, with careful endplate preparation via curetting to avoid excessively damaging the endplate and maintaining geometry. Placement of the PSI vertebrectomy device was performed with four integral fixation screws (16 mm length—as determined by the VSP) with a mixture of autograft and allograft within the central cage graft void. Radiography was used to confirm final placement. Copious antibiotic irrigation was used. Further bone graft was packed around the cage. Fascial and skin layers were closed. Throughout the procedure, the 3DP biomodel of the patient’s spine and the custom cage were referred to as needed.

## 3. Results

Intraoperatively, the implant exhibited excellent fit and primary stabilization. The surgical team noted the biomodels (opaque dynamic and clear C7) allowed identification of the position and course of the vertebral arteries and nerve roots (Figure 1C). These, combined with VSP and PSIs, were used to determine the (width) extents of the corpectomy, enabling the corpectomy-cord decompression part of the procedure to be performed rapidly and precisely. The pre-planned screw trajectories and lengths also minimised the time and radiographic imaging needed during screw fixation of the implant. Immediate postoperative imaging confirmed satisfactory decompression of the cord and instrumentation positioning. The patient was extubated on Day 1 post-surgery and transferred to the spinal injuries rehabilitation unit. The neurological deficit recovered rapidly with restoration of anti-gravity movement within 72 h post-operation. The patient returned to independence within three weeks and was discharged by week four; however, with a significant dysfunction of gait metrics including a mean walking speed of 0.78 m/s, step length of 52 cm and daily step count of <1000/day. At three months follow-up, the patient had significantly improved all objective gait parameters (walking speed 1.07 m/s, step length 64 cm and average daily step count 2600/day), and the radiographic outcome remained excellent with (fusion) bone observed through the length of the PSI and minimal implant subsidence and/or migration (Figure 2, Figure 3). At 15 months follow-up, the patient was living independently with minimal restriction of range of motion and no neck pain and returned to normal life without ongoing medical care necessary. Objective walking metrics had returned to normal parameters [15,16] for a male of his age with a walking speed of 1.31 m/s, step length of 72 cm and average daily step count of >10,000/day.

## 4. Discussion

The authors report the first application of a custom-made prosthesis to an emergency trauma case. The C7 burst fracture, with associated spinal cord compression, resulting from acute axial compression loading presented here was successfully treated with a rapidly produced PSI. The case demonstrates how patient transport and acute care as well as device manufacture can be run in parallel and temporo-spatially converge to optimise care for an emergency patient. While demonstrated here in a spinal patient, the broader approach can readily be modified and applied to other emergency trauma. As such, this report adds to the growing literature describing patient-specific approaches to orthopaedic trauma [17,18].

Following imaging of the pathological anatomy, VSP optimises various aspects of the surgical workflow including: the extent of bone removal for effective decompression; distractor (e.g., Caspar pin) placement and implant placement (including using pre-planned screw lengths and trajectories). The VSP process combined with postoperative imaging also allows for quantification of reconstruction compared with the surgical plan. The VSP combined with PSI reduces the need for intra-operative surgical ‘improvisation’, trial and error or stepwise fixation, which may reduce the risk of complications whilst maximizing the stabilization achieved (through the use of optimal screw length and trajectory).

A key feature of the personalised approach to spinal surgery is the matching of the cage dimensions and surface topology to the patient’s unique endplate morphology, allowing for an implant with a patient-specific, maximised footprint. This is advantageous given research highlighting the importance of cage footprint and maximised implant-bone contact area for osseointegration, even stress distribution between device and vertebral bodies and in reducing cage subsidence [3,18,19]. Contoured device–anatomy interfaces on implants may also help minimise endplate preparation, allowing for preservation of stiffer endplate bone and preventing direct loading of underlying cancellous bone, as well as allowing for stronger peripheral endplate bone to be loaded by the cage, again minimizing the risk of cage subsidence [20,21,22,23,24].

This case presents a strong argument for rapid delivery of on-demand PSIs to the ‘point-of-care’. This is particularly valuable in trauma surgery where there may be irregular and complex bony surfaces following fracture and/or dislocation, and earlier surgery may improve patient and institutional outcomes [25]. In contrast, the conventional paradigm involves delaying definitive fixation until implants and access instruments are available, and the patient is more able to tolerate physiologically taxing surgery [26]. This on-demand personalised approach to trauma management may help solve issues of early surgical intervention. If an early procedure is required nonetheless, for example, to decompress the spinal canal [25], then, if feasible, this technology would allow this opportunity to be capitalised on for definitive management.

While there has been growing interest in the prospect of ‘at-the-point-of-care’ providers, including major hospital complexes having their own additive manufacturing capacity, significant concerns remain regarding the complexities of costly and difficult setup and implementation, manufacturing efficiencies and regulatory obligations. An alternative model which helps address these issues, and is demonstrated by this case, is the establishment of ’near-the-point-of-care’ facilities which service multiple nearby hospitals, thereby reducing overall costs and concentrating specialised skills.

Ultimately, the question of where personalised devices are manufactured will likely continue to be debated and come down to cost, quality, regulations and speed of implant delivery. Increased costs due to each custom device requiring design input and reduced economies of scale due to smaller production batches may partially be offset by the increased efficiency of additive manufacturing over a subtractive approach, with reduced raw material wastage and inventory costs (storage, logistics, expiry-wastage). While the regulatory frameworks for non-‘at-the-point-of-care’ manufacturers have now been defined in many jurisdictions, the regulations for ‘at-the-point-of-care’ production are still being developed, leading to uncertainty and delay in the implementation of these types of production facilities. Balanced regulation can be complex and time consuming to achieve, with industry and the relevant regulatory bodies agreeing and accepting a transition-of-care model to a personalised approach.

As demonstrated by this case, advances in PSI design and 3DP manufacturing processes have enabled the application of these technologies, with their associated benefits, to emergency trauma cases. This approach may represent the next phase of efficient surgical planning and workflow, as well as the possible future of personalised healthcare in trauma, orthopaedics and reconstructive surgery.

## Figures and Tables

**Figure 1 jpm-12-00997-f001:**
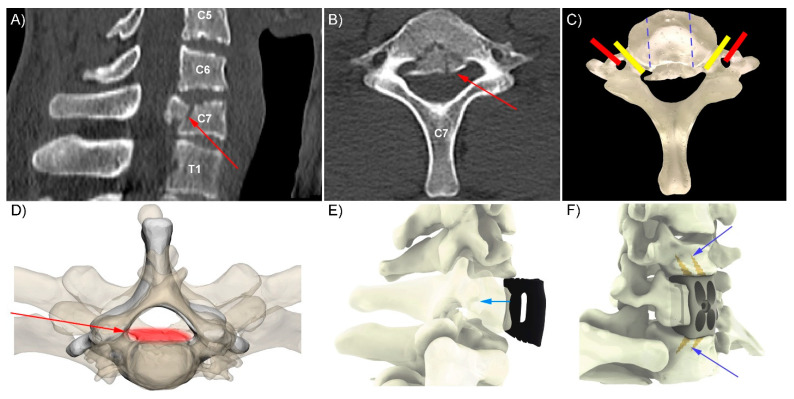
Rapid preoperative design and production. Preoperative imaging received from peripheral hospital demonstrated, via sagittal (**A**) and axial (**B**) bone windows, a C7 burst fracture with retropulsion of fragment (red arrows in **A**,**B**,**D**) into the spinal canal with stenosis (**A**–**D**). Production of a 3D-printed biomodel (**C**) and anterior approach patient-specific implant design (**E**) for use with integral fixation screws (**F**, blue arrows). The biomodel (**C**), combined with the 3D-printed implants (see Figure 2) aided determination of the width of vertebral body resected (blue dashed lines); yellow shows nerve root paths, red shows location of vertebral arteries.

**Figure 2 jpm-12-00997-f002:**
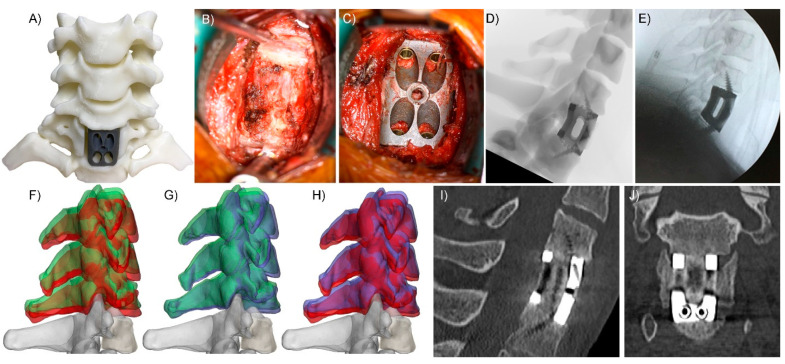
Implantation and postoperative imaging. (**A**) dynamic biomodel allowed for trialling of different sized implants and aided in determining the extents of C7 vertebral body resection. Surgical decompression prior to implant placement. (**B**) Immediate press fit of the implant was achieved with integral screw fixation used to maximise initial stability of the construct (**C**). Virtual surgery planning (VSP) simulated X-rays (**D**) were compared to intra-operative X-rays (**E**) to check depth of implantation and screw trajectories. (**F**) C7 (grey) with pathological (red) C4-6 vertebral relative positioning compared to immediate postoperative positioning (green) of the same vertebrae. Note that the procedure restored height (green [post-op] is higher than red [pre-op]). (**G**) Green, as in F, is the immediate postoperative vertebrae positioning compared to blue, which is 10 weeks post-op. (**H**) Ten-week post-op (blue) vertebral positioning compared to pre-op pathological (red) positioning. Ten-week post-op sagittal (**I**) and coronal (**J**) CT slices demonstrate: excellent cord decompression, new (fusion) bone growth through the graft window of the device, the stability of the construct (osseointegration with the device), excellent implant positioning.

**Figure 3 jpm-12-00997-f003:**
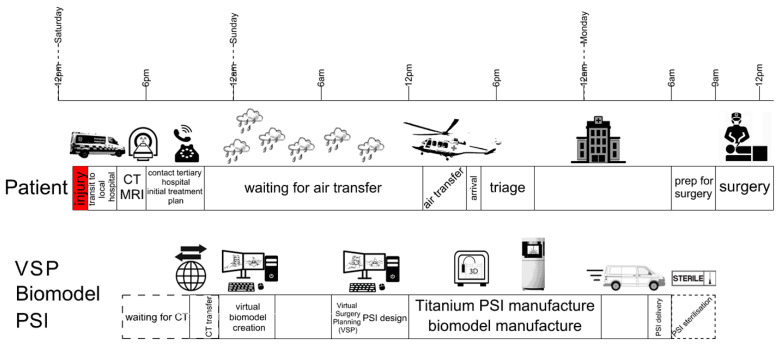
Chronology of rapid implementation of personalised spinal trauma surgery. Patient and device manufacture timelines. The patient was injured in a rural location shortly after midday on Saturday and was transported to a local hospital for medical imaging. Primary diagnosis led to contact with a tertiary hospital with a spinal unit and transfer of the computer tomography imaging, which was used for virtual surgical planning, biomodel and patient-specific implant (PSI) design. After a weather delay, the patient was airlifted to the tertiary hospital. Meanwhile, the biomodel and PSI were being manufactured at a ‘near-the-point-of-care’ facility. After manufacture, post-processing and quality checks, the implants were delivered and sterilised by the hospital’s Central Sterile Supply Department. Surgery was scheduled as first on Monday’s list.

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
