# Peer review of "Rapid Personalised Virtual Planning and On-Demand Surgery for Acute Spinal Trauma Using 3D-Printing, Biomodelling and Patient-Specific Implant Manufacture"

_jpm, 2022, doi:10.3390/jpm12060997_

Round 1

Reviewer 1 Report

I read this article with pleasure, the case report is really interesting and confirms that image acquisition and segmentation, 3D modeling, VSP, and  3D-printing of PSIs can be safely and effectively completed in a very short time, by close collaboration between surgeons and CAD engineers and "near-the-point-of-care" facilities.

Author Response

Dear reviewer.

Thankyou for your kind comments on our paper! There are no major changes as per your reviewe comments. The report form highlighted "english language and style" and this will be addressed in the manusciprt reply.

With thanks,

Ralph Mobbs

Reviewer 2 Report

The article is well written and the presentation of the case is well documented with good quality pictures.

Authors adequately described the personalized Patient-Specific-Implant including the timing for obtaining PSI manufacture biomodel.

In this type of trauma and generally speaking during emergency, the organization and the timing of preparation of PSI are the key for good outcomes and these aspects have been underlined, together with the problem of costs, implant delivery and quality of manufacture. 

Minor revisions:

Page 6 “In the present case, a fracture at the anterior margin of the C6 vertebral body (superior) was addressed with the custom implant being designed to avoid loading this region, thereby aiming to minimize the risk of the C6 fracture propagating or a fragment of C6 breaking loose post-operatively.”: the fracture of anterior c6 is not mentioned and decribed in the case or visible in figure

In references’ list, DOI is lacking in paper n. 1 and n. 18

Author Response

Dear Reviewer,

Thankyou for your kind comments and comprehensive review.

There are several issues highlighted:

  1. C6 fracture text. Yes! On review, we agree that this paragraph adds nothing to the article and the fracture is not seen on the provided imaging. This fracture was negligable on the preopertive imaging, and we have decided to remove this from the text.
  2. Missing DOI. This has been added to the revised manuscript submitted.

Thankyou for your time reviewing our article,

Ralph Mobbs

This manuscript is a resubmission of an earlier submission. The following is a list of the peer review reports and author responses from that submission.